# Chinese Traditional Fermented Soy Sauce Exerts Protective Effects against High-Fat and High-Salt Diet-Induced Hypertension in Sprague-Dawley Rats by Improving Adipogenesis and Renin-Angiotensin-Aldosterone System Activity

**Bao Zhong [1,2], Eun-Gyung Mun [1] , Jin-Xi Wang [1] and Youn-Soo Cha [1,*]**

[1] Department of Food Science and Human Nutrition, Jeonbuk National University and Obesity Research Center, Jeonju-si 54896, Korea; zhongbao19870626@163.com (B.Z.); egmun1982@gmail.com (E.-G.M.); jinxiwang3@gmail.com (J.-X.W.)

[2] School of Food Engineering, Jilin Agriculture Science and Technology University, Hanlin Road, Changyi District, Jilin 132000, China

* Correspondence: cha8@jbnu.ac.kr; Tel.: +82-63-270-3822

**Abstract:** Although high-fat and high-salt diets are considered risk factors for hypertension, the intake of salty soybean-based fermented foods has beneficial effects. This study explored the potential of Chinese traditional fermented soy sauce (CTFSS) in preventing hypertension by analyzing its effects on adipogenesis and the renin-angiotensin-aldosterone system (RAAS). Male Sprague-Dawley (SD) rats were divided into four groups ($n = 6$): normal diet (ND), high-fat diet (HD), high-fat diet with saline (HDS, NaCl-8%), and high-fat diet with Chinese traditional soy sauce (HDCTS, NaCl-8%). Each group is administered 12 weeks by oral gavage as 10 mL/kg dose, respectively. CTFSS supplementation resulted in significantly lower body weight, epididymal fat weight, and systolic blood pressure. Additionally, it decreased the serum total cholesterol (TC), triglyceride (TG), alanine aminotransferase (ALT), aspartate aminotransferase (AST), renin, angiotensin II (Ang II), angiotensin-converting enzyme (ACE), and aldosterone levels. It also increased the urinary volume and improved sodium and potassium ion balance. The gene levels showed significant enhancements in the mRNA levels of renin-angiotensin-aldosterone system-related and adipogenesis-related genes. In addition, CTFSS may prevent hypertension-associated kidney injury. Therefore, this study demonstrates that CTFSS has no harmful effects on hypertension. In contrast, the beneficial effects of CTFSS intake in ameliorating hypertension were shown.

**Keywords:** soy sauce; salt; hypertension; adipogenesis; renin-angiotensin-aldosterone system

## 1. Introduction

Hypertension is a growing public health challenge worldwide. It affects approximately 40% of the world population over the age of 25, and by 2025, the global prevalence of hypertension will increase by 60%, and cardiovascular disease (CVD) is becoming an increasingly common health problem worldwide [1]. Hypertension can induce more than half of the coronary heart disease burden, leading to premature death [2]. A high-salt and high-fat diet is an important risk factor for hypertension, atherosclerosis, and coronary heart disease [3]. The World Health Organization (WHO) recommends less than 5 g/day salt intake, whereas the Chinese Nutrition Society (CNS) recommends a salt intake of less than 6 g/day. However, the daily salt intake in Chinese population is approximately 10.5 g/day. Long-term intake of large amounts of sodium can easily lead to the retention of sodium and water, leading to increased blood pressure [4]. Obesity is a metabolic syndrome, which accounts for hypertension in approximately 72% of men and women [5].

High-fat diet may increase lipid levels and promote angiotensinogen (AGT) production by adipocytes, leading to renin-angiotensin-aldosterone system (RAAS) activation [6,7]. High-fat diet-fed mice showed increased expression of angiotensin II (Ang II), angiotensin-converting enzyme (ACE), and aldosterone in the kidney, indicating increased RAAS activity [8]. High sodium intake leads to RAAS dysfunction, and hypertension is associated with activated RAAS [9]. High salt sensitivity has been shown to lead to obesity and vascular dysfunction via RAAS activation [10]. Salt increases the metabolites involved in the RAAS, such as aldosterone synthase (Cyp11b2) and mineralocorticoid receptor (MR), and promotes rising serum aldosterone levels, sodium retention and volume expansion, renal disease, and elevated blood pressure [11,12]. High-salt diets accelerate hypertension and renal inflammation/injury in male rats such as low-grade renal histological injury, glomerular hyaline casts, interstitial fibrosis (peritubular), glomerular sclerosis, tubular atrophy [13].

Soy sauce originated in China over 2500 years ago. Presently, soybean-fermented foods are popular worldwide, especially in most East Asian countries [14]. Soybean fermentation can increase the content of isoflavones in aglycones, thereby increasing the bioavailability of isoflavones [15]. Koji is made from wheat and soybeans fermented with *Aspergillus oryzae* and is essential for the production of Chinese traditional fermentation soy sauce (CTFSS) [16]. A placebo-controlled double-blind trial showed that supplementation with *koji* decreased systolic and diastolic pressure significantly in patients with hypertension [17]. Other than just a seasoning, soy sauce is also a potential functional food with anti-obesity and anti-hypertensive effects. In previous studies, soy sauce intake was found to alter the fat metabolism of *Caenorhabditis elegans* [18]. Traditional Korean soy sauce can ameliorate hypertension in rats [19].

The RAAS is a hormonal cascade that functions in the homeostatic control of arterial pressure, and the imbalance of the RAAS is crucial in the pathogenesis of cardiovascular and renal disorders [20]. Salty fermented soybean foods, such as Japanese soy sauce, can effectively inhibit ACE activity, thereby regulating blood pressure [21]. Korean soy sauce is manufactured using meju, unlike its Chinese counterpart. Although there are differences in the manufacturing methods of Chinese soy sauce and Korean soy sauce, the fermentation bacteria are similar, and all the fermented metabolites contain lactic acid bacteria and *Bacillus* [22,23]. Chinese soy sauce and Japanese soy sauce were prepared using the same method [24]. However, consumption of Japanese soy sauce does not increase the blood pressure [25]. Most studies on Chinese soy sauce have focused on analysis of aroma and composition content. However, research on the functionality of Chinese fermented soy sauce is limited. We hypothesized that the intake of Chinese traditional soybean-fermented foods has a regulatory effect on high-fat and high-salt diet-induced hypertension. Therefore, in this study, the antihypertensive effects of CTFSS were investigated in Sprague-Dawley (SD) rats.

## 2. Results

### 2.1. Metabolic Characterization, $Na^+$ and $K^+$ Ion Concentration, and Serum Chemistry

The results are shown in Table 1. The initial body weight was insignificantly different between all the groups. The final body weight significantly decreased in the HDCTS group compared to that in the HD and HDS groups. The epididymal fat weight of the HDCTS group was significantly lower than that of the HD and HDS groups. Diet intake in ND, HD, and HDCTS groups compared to the HDS group were significantly different, while same was insignificantly different between the ND, HD, and HDCTS groups. Results of the 24-h metabolic cage experiment showed that urine metabolism significantly increased in the HDCTS group compared to the other groups. No significant difference in fecal excretion in all groups. Water intake was significantly increased in the HDS and HDCTS groups than in the ND and HD groups.

**Table 1.** Metabolic characterization, $Na^+$ and $K^+$ ion concentration, and serum chemistry.

| Group | | ND | HD | HDS | HDCTS |
|---|---|---|---|---|---|
| Metabolic characterization | | | | | |
| Initial body weight (g) | | $116.23 \pm 3.18$ | $116.98 \pm 8.33$ | $117.22 \pm 7.06$ | $116.12 \pm 5.63$ |
| Final body weight (g) | | $454.52 \pm 6.94$ [c] | $556.69 \pm 28.14$ [a] | $547.65 \pm 33.82$ [ab] | $534.98 \pm 35.78$ [b] |
| Epididymal fat weight (g)/BW (g)% | | $1.84 \pm 0.25$ [c] | $3.64 \pm 0.48$ [a] | $3.46 \pm 0.26$ [a] | $2.82 \pm 0.38$ [b] |
| Food intake (g/day) | | $17.22 \pm 0.19$ [a] | $17.05 \pm 0.10$ [a] | $16.51 \pm 0.39$ [b] | $17.14 \pm 0.07$ [a] |
| Water intake (mL/day) | | $16.33 \pm 1.18$ [c] | $17.71 \pm 1.48$ [c] | $25.08 \pm 2.16$ [a] | $21.65 \pm 2.50$ [b] |
| Urinary volume (mL/day) | | $9.58 \pm 1.88$ [c] | $9.63 \pm 0.77$ [c] | $13.50 \pm 1.22$ [b] | $15.08 \pm 0.66$ [a] |
| Fecal excretion (g/day) | | $3.31 \pm 0.14$ | $3.24 \pm 0.15$ | $3.16 \pm 0.19$ | $3.32 \pm 0.14$ |
| $Na^+$ and $K^+$ ion concentration in urine and feces (ppm) | | | | | |
| Urine | $Na^+$ | $2130.33 \pm 698.16$ [b] | $1844.92 \pm 620.88$ [b] | $9314.08 \pm 978.00$ [a] | $8778.68 \pm 354.74$ [a] |
| | $K^+$ | $9292.65 \pm 2594.60$ [ab] | $10,084.34 \pm 1359.43$ [a] | $6837.20 \pm 459.40$ [b] | $7046.87 \pm 1079.26$ [ab] |
| Feces | $Na^+$ | $722.82 \pm 317.74$ | $796.94 \pm 339.65$ | $1116.71 \pm 200.52$ | $603.91 \pm 283.51$ |
| | $K^+$ | $6387.93 \pm 1357.67$ | $6232.49 \pm 756.63$ | $4734.04 \pm 246.97$ | $4795.09 \pm 708.70$ |
| Serum chemistry | | | | | |
| TC (mg/dL) | | $66.44 \pm 6.09$ [c] | $95.33 \pm 4.35$ [a] | $91.16 \pm 8.97$ [a] | $78.28 \pm 5.83$ [b] |
| TG (mg/dL) | | $51.52 \pm 4.19$ [d] | $116.91 \pm 4.66$ [a] | $97.83 \pm 5.90$ [b] | $60.44 \pm 7.03$ [c] |
| AST (IU/L) | | $39.41 \pm 2.43$ [d] | $56.42 \pm 2.93$ [a] | $52.59 \pm 1.70$ [b] | $42.68 \pm 1.29$ [c] |
| ALT(IU/L) | | $11.10 \pm 1.28$ [b] | $18.66 \pm 1.65$ [a] | $17.70 \pm 0.74$ [a] | $11.34 \pm 1.67$ [b] |

Values are shown as the means $\pm$ standard deviation and statistical difference from each other at $p < 0.05$ by Duncan's multiple range test (a > b > c > d). ND: normal diet; HD: high-fat diet; HDS: high-fat diet + saline; HDCTS: high-fat diet + Chinese traditional soy sauce.

Compared to the ND and HD groups, urine concentration of $Na^+$ was significantly increased in the HDS group, whereas the HDCTS group showed a downward trend in the concentration of urine $Na^+$ compared to the HDS group, although this difference was statistically insignificant. The concentration of urine $K^+$ was significantly higher in the HDCTS group than in the HDS group. In feces, the HDCTS group when compared with the HDS group shows the $Na^+$ and $K^+$ to be statistically insignificant.

In HD and HDS groups showed significantly higher levels of serum TC, TG, AST, and ALT compared to the HDCTS group.

### 2.2. Systolic Blood Pressure

The alterations in systolic blood pressure (SBP) are shown in Figure 1. The initial SBP did not differ significantly between the groups. At six weeks of experiments, in the HDCTS group, blood pressure was significantly attenuated compared to the HD and HDS groups. SBP was significantly higher in the HD and HDS groups than in the other groups at 12 weeks. However, the final SBP was significantly attenuated in the HDCTS group. (Supplementary Figure S1).

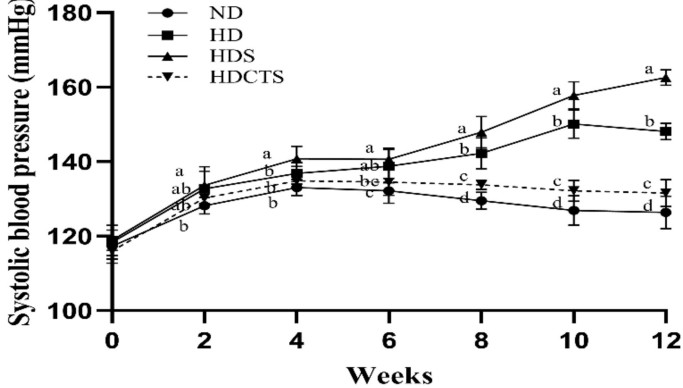

**Figure 1.** Changes in SBP during the experiment. Values are shown as the means $\pm$ standard deviation, and statistical differences are expressed in letters (a > b > c > d). ND: normal diet; HD: high-fat diet; HDS: high-fat diet + saline; HDCTS: high-fat diet + Chinese traditional soy sauce.

### 2.3. Renin, Ang II, ACE, and Aldosterone Levels in Serum

The test results are shown in Figure 2. The serum levels of renin, Ang II, ACE, and aldosterone were significantly higher in the HD and HDS groups than in the ND group, and a significant increase was observed in the HDCTS group.

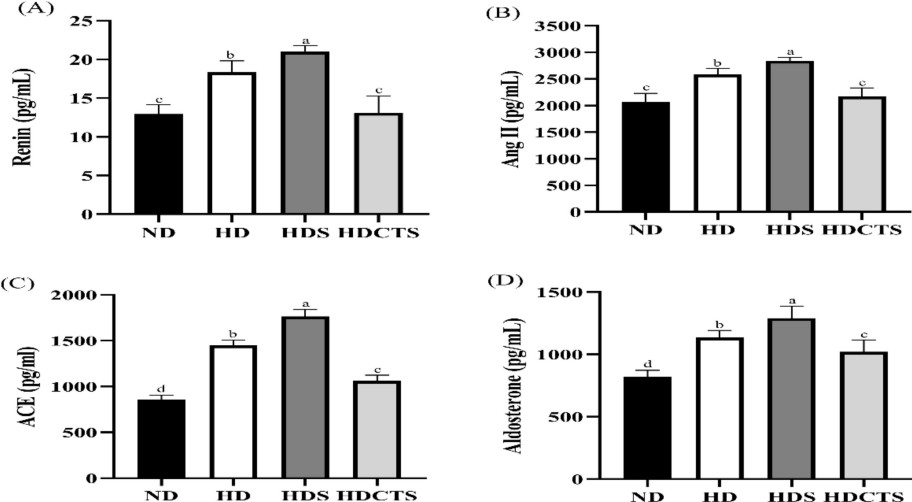

**Figure 2.** Renin-angiotensin-aldosterone relative levels in serum. (**A**) Renin level of serum; (**B**) Ang II level of serum; (**C**) ACE level of serum; (**D**) Aldosterone level of serum. Values are shown as the means ± standard deviation. Values with different superscripts letters (a > b > c > d) are significantly different among groups. ND: normal diet; HD: high-fat diet; HDS: high-fat diet + saline; HDCTS: high-fat diet + Chinese traditional soy sauce.

### 2.4. Histopathology of Kidney

Histological examination confirmed significant glomerular hypertrophy and slight hyaline degeneration in the kidney sections from rats in the HD and HDS groups compared to the ND group. The glomerular area and hyaline degeneration in the HDCTS group were significantly lower than the HD and HDS groups (Figure 3).

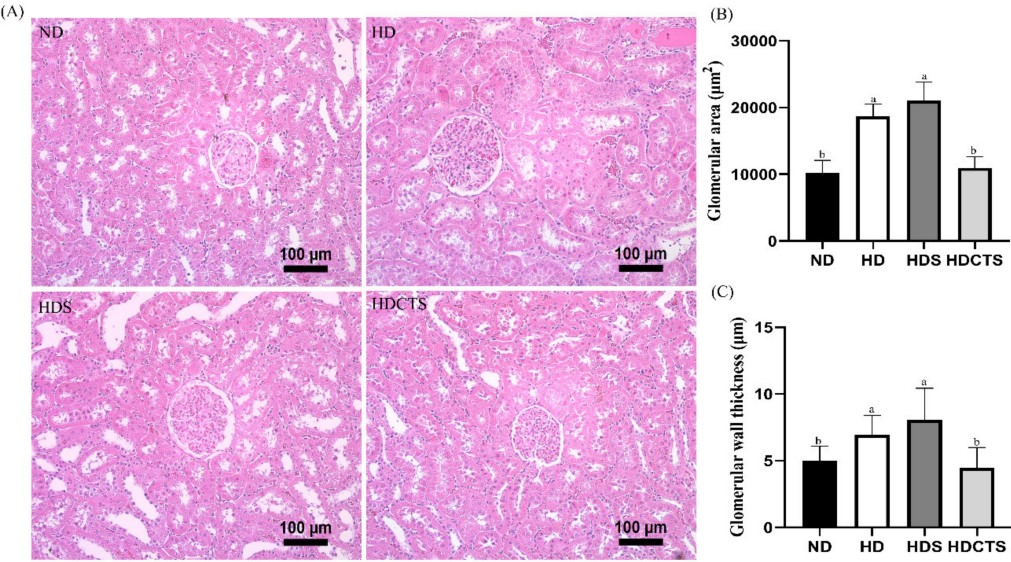

**Figure 3.** Histological analysis of kidney tissue. (**A**) Histological examination (20×) in kidney; (**B**) Glomerular area; and (**C**) Glomerular wall thickness. Values are shown as the means ± standard deviation. Values with different superscripts letters (a > b) are significantly different among groups. ND: normal diet, HD: high-fat diet, HDS: high-fat diet + saline, and HDCTS: high-fat diet + Chinese traditional soy sauce.

### 2.5. Expression of Adipogenesis-Regulating and RAAS-Related Genes in Liver Tissues

The mRNA expression of adipogenesis-related genes, leptin and PPARγ, significantly increased in the HD and HDS groups compared to the ND group. However, the levels of these genes in the HDCTS group were lower than those in the HD and HDS groups. The mRNA level of adiponectin was significantly higher in the HDCTS group than in the HD and HDS groups. The expression of the RAAS-related gene AGT was significantly increased in the HD and HDS groups compared to the ND group, and was downregulated in the HDCTS group (Figure 4).

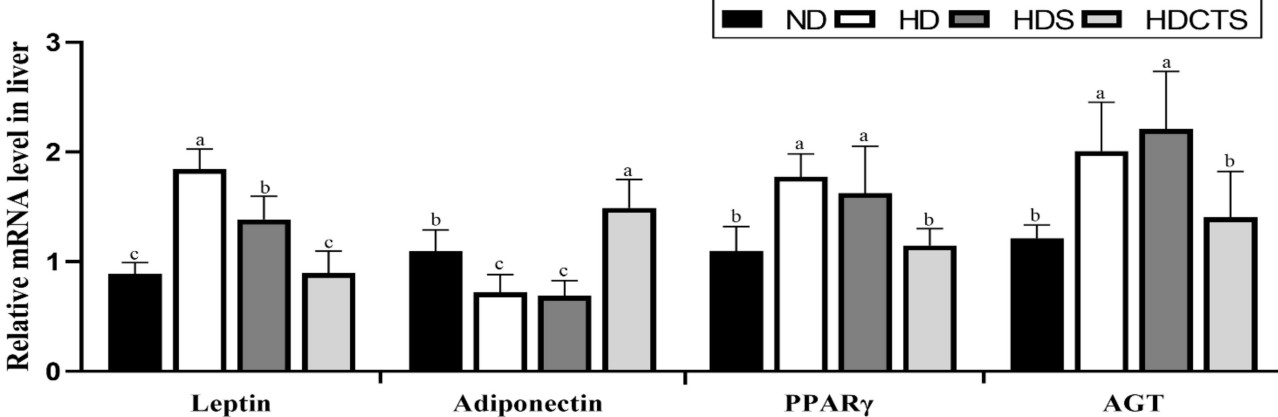

**Figure 4.** The mRNA expression of adipogenesis- and RAAS-related genes in the liver tissue. Values are shown as the means ± standard deviation. Values with different superscripts letters (a > b > c) are significantly different among groups. ND: normal diet; HD: high-fat diet; HDS: high-fat diet + saline; HDCTS: high-fat diet + Chinese traditional soy sauce; PPARγ: Peroxisome proliferator-activated receptor gamma; AGT: Angiotensinogen.

### 2.6. Expression of RAAS-Related Genes in Kidney Tissues

The mRNA expression of RAAS-related genes, renin, ACE, and AT1 was significantly overexpressed in the HD and HDS groups compared to the ND group, whereas the HDCTS group significantly downregulated the expression levels of these genes. In the HD and HDS groups, Cyp11a1, Cyp11b2, Hsd3b1, Star, and MR levels were significantly increased compared to the ND group, and in the HDCTS group, these genes were significantly decreased. Notably, the expression of Rnls was significantly reduced in the HD and HDS groups compared to the ND group, and increased in the HDCTS group (Figure 5).

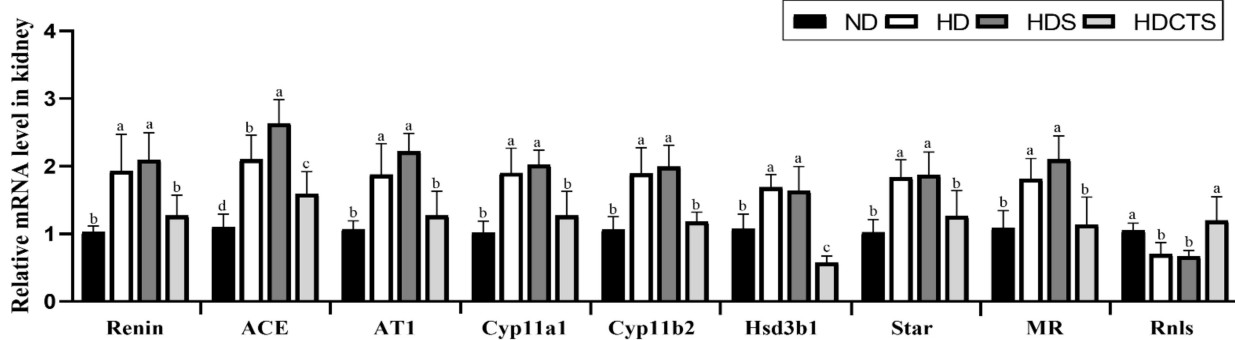

**Figure 5.** The RAAS-related mRNA expression levels in the kidney tissue. Values are shown as the means ± standard deviation. Values with different superscripts letters (a > b > c > d) are significantly different among groups. ND: normal diet; HD: high-fat diet; HDS: high-fat diet + saline; HDCTS: high-fat diet + Chinese traditional soy sauce; ACE: Angiotensin-converting enzyme; AT1: Angiotensin type 1 receptor; Cyp11a1: Cholesterol side-chain cleavage enzyme; Cyp11b2: Aldosterone synthase; Hsd3b1: 3β-hydroxysteroid dehydrogenase type 1; Star: Steroidogenic acute regulatory protein; MR: Mineralocorticoid receptor; Rnls: Renalase.

## 3. Discussion

The occurrence of high blood pressure is related to changes in lifestyle, such as increase sodium intake, decreased potassium intake, and decreased vegetarian diet [26]. High-fat diet consumption, which causes increases the risk of hypertension [27]. The high sodium intake and the increase in SBP levels are related to water and sodium retention and modification of sympathetic activity [28]. Some evidence suggests that soybean-fermented foods, such as Korean paste, soy sauce, and Japanese miso, can regulate obesity and blood pressure [19,29,30]. In this study, we demonstrated that the intake of CTFSS for 12 weeks prevented body weight gain, epididymal fat weight gain, and increased blood pressure.

Salty soybean fermentation food supplementation is related to decreases in serum lipid levels, which may be partially due to the higher content of soybean isoflavones in soybean fermentation food [31,32]. In this study, the serum levels of TC, TG, ALT, and AST levels significantly decreased in the HDCTS group than in the HD and HDS groups. The amount of renin in the serum is the key rate-limiting step determining the level of ACE [33]. Increased aldosterone levels in serum are a risk of hypertension [34]. In this study, CTFSS supplementation significantly lowered the expression of serum renin, ACE, Ang II, and aldosterone.

Early manifestations of renal injury caused by hypertension include structural changes, such as glomerular hypertrophy, followed by thickening of the glomerular walls and hyaline degeneration [13]. Numerous studies have demonstrated that Ang II induces glomerular hypertrophy and glomerulosclerosis by activating its specific receptor [35]. In the present study, significant hypertrophy and slight hyaline degeneration of glomeruli were observed in the HD and HDS groups. However, significant prevention of glomerular hypertrophy and hyaline degeneration was observed in the HDCTS group. Therefore, in this study, we demonstrated that CTFSS may prevent hypertension-associated renal injury.

High leptin and low adiponectin levels are characteristic of obesity. PPARγ plays a vital role in the early stage of adipose differentiation [36]. In this study, CTFSS suppressed leptin and PPARγ gene expression, and increased adiponectin gene levels. Therefore, the results of our study showed that CTFSS exerts lipid-lowering effects. Liver-derived AGT is the precursor to all angiotensin and is converted via renin to form angiotensin I (Ang I) and induces RAAS activation, thereby increasing blood pressure [37]. Previous studies have shown that the expression level of AGT in obese mice increased significantly, and AGT knockout showed lower blood pressure regulation [38]. Increased leptin and PPARγ can modulate the local production of AGT, thereby increasing blood pressure by activating the RAAS [39]. Decreased adiponectin expression causes an increase in angiotensinogen, thereby increasing blood pressure [40]. Salt intake has an inappropriate augmentation of AGT, which may contribute to hypertension [41]. In this study, AGT levels were significantly increased in the HD and HDS groups compared to the ND group, and a significant downregulation of AGT levels was observed in the HDCTS group.

The renin activity may be a marker of CVD risk in hypertensive patients [42]. ACE can generate the vasoactive peptide Ang II by cleaving 2 amino acids from the C-terminus of the inactive precursor Ang I [43]. Obese subjects, significant increases in ACE activity, and Ang II, thereby induced elevated SBP [44,45]. High-salt diet-induced hypertension in rats and renin and ACE levels increased significantly [46]. However, miso sauce intake has potent angiotensin-converting enzyme inhibitory effects and can reduce nighttime blood pressure [47]. In this study, CTFSS supplementation decreased the gene expression of renin and ACE.

The high-salt diet stimulates glomerular oxidative stress, which leads to AT1 receptor upregulation, subsequently causing sodium retention and hypertension [48]. Ang II may activate Cyp11b2 and Star expression and promote aldosterone secretion [49,50]. Cyp11a1 catalyzes the side-chain cleavage of cholesterol, and Cyp11b2 catalyzes the final steps in the biosynthesis of MR [51]. Increased levels of Hsd3b1 and Star genes may upregulate blood pressure via changes in aldosterone [52,53]. Furthermore, MR-mediated changes in the cardiovascular system are potentiated by activation of AT1 receptors [54]. A previous

study reported that Korean soy sauce might decrease AT1, aldosterone, and MR levels, and downregulate the expression of $Na^+/K^+$ ATPase$\alpha$1, and improve $Na^+$ reabsorption, thereby decreasing blood pressure [19]. The present study demonstrated that CTFSS supplementation significantly decreased the mRNA levels of AT1, Cyp11a1, Cyp11b2, Hsd3b1, Star, and MR. In the HDCTS group, the balance of sodium and potassium ions also showed an adjustment, and urine volume increased significantly.

Previous studies have shown that sympathetic nervous system activity is heightened by the activation of brain regions controlling autonomic function due to a high-fat diet, salt, and Ang II [55]. Inappropriate activation of intrarenal RAAS may not contribute directly to the decreased expression of renalase (Rnls) in the rats that received a high-salt diet. However, activated RAAS promotes sympathetic nervous activation, which may decrease the renal Rnls expression [56]. Our research found that the expression level of Rnls was significantly reduced in the HD and HDS groups and increased significantly in the HDCTS group. In the present study, the results suggest that the consumption of CTFSS may have a lower risk of hypertension (Figure 6).

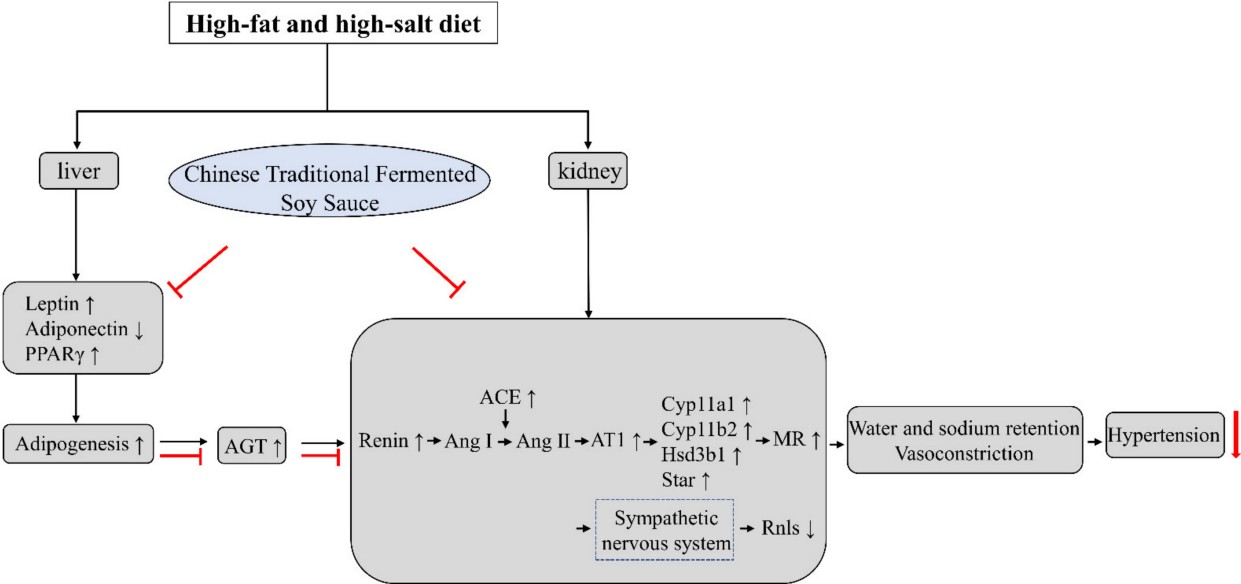

**Figure 6.** Mechanism underlying the hypertension-regulating action of CTFSS. When compared to high-fat and high-salt intake, adipogenesis and RAAS-relative levels were regulated with CTFSS. PPAR$\gamma$: Peroxisome proliferator-activated receptor gamma; AGT: Angiotensinogen; Ang I: Angiotensin I; Ang II: Angiotensin II; ACE: Angiotensin-converting enzyme; AT1: Angiotensin type 1 receptor; Cyp11a1: Cholesterol side-chain cleavage enzyme; Cyp11b2: Aldosterone synthase; Hsd3b1: 3$\beta$-hydroxysteroid dehydrogenase type 1; Star: Steroidogenic acute regulatory protein; MR: Mineralocorticoid receptor; Rnls: Renalase. "T" bars: The inhibitory effect of Chinese traditional fermented soy sauce.

Chinese soybean fermented foods, such as soybean paste and soy sauce, are important sources of salt intake [57,58]. The intake of a high-fat and high-salt diet is considered a risk factor for hypertension [28,59]. However, the present study showed that CTFSS has no harmful effects, such as elevated blood pressure. In contrast, the intake of CTFSS in rats fed with a high-fat and high-salt diet might alleviate hypertension.

## 4. Materials and Methods

### 4.1. Preparation of Soy Sauce

The CTFSS was supplied by JiangCheng Brewing Group Co., Ltd. (Jilin, China). Soybeans were washed and soaked overnight, then steamed at 121 °C for 10 min, cooled to room temperature, mixed with bran-free raw wheat flour, inoculated with *Aspergillus oryzae*, and kept at 30 °C for 48 h. Koji was mixed with 2.5 times volume of brine solution, fermented and separated into liquid phases, and soy sauce was harvested (Figure 7). The

salinity of the soy sauce was adjusted to 8% using distilled water by analysis using Mohr's method [60].

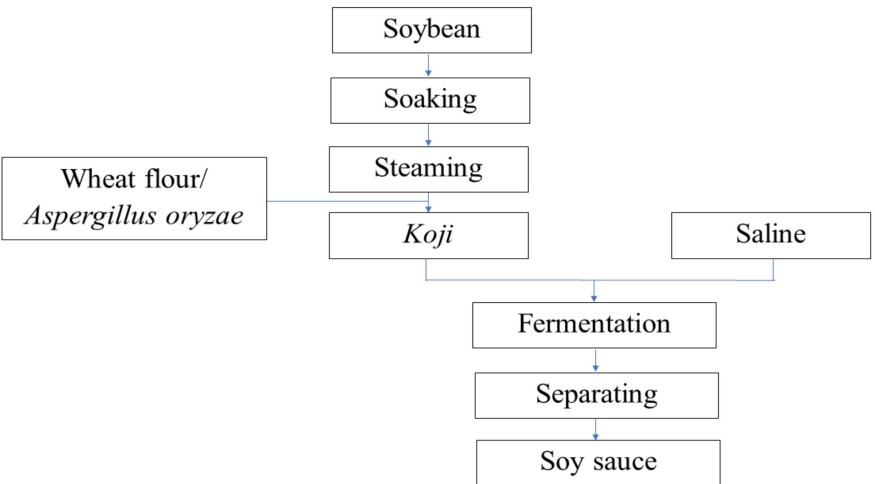

**Figure 7.** Procedure for manufacturing Chinese traditional fermented soy sauce.

### 4.2. Animal Experiment

Twenty-four male, five-week-old SD rats were purchased from DooYeol Biotech (Seoul, Korea). After adaptation for 1 week, SD rats were divided into four groups (*n* = 6) by non-significantly different blood pressure and body weight: normal diet (ND), high-fat diet (HD), high-fat diet and saline (HDS, NaCl-8%), and high-fat diet and Chinese traditional soy sauce (HDCTS, NaCl-8%). Wooden chips (JRS, Rosenberg, Germany) were used as bedding in the cages. The animal room temperature was maintained at $23 \pm 1\,^\circ$C, and the relative humidity was maintained at $65 \pm 5$%. The lighting was adjusted at 12 h light/dark cycle. The rats had free access to food and tap water. Diet intake was recorded every 2 d. All the rats were orally gavage administered by gastric intubation for 12 weeks, with the dose 10 mL/kg body weight. (Supplementary Figure S2).

### 4.3. Measurement of Body Weight and Blood Pressure

During feeding, body weight was measured once a week, and blood pressure was measured every 2 weeks using the tail-cuff method (BP-2000 series II blood pressure analysis system, Apex, NC, USA) after 3 h of oral gavage.

### 4.4. Metabolic Cages Experiment and Urine and Feces Analysis

During the last 4 weeks of the experiment, rats were kept in metabolic cages for 24 h a week, all feces and urine were collected, and water intake was recorded. Concentrations of sodium and potassium ions in urine and feces were analyzed at the Center for University-wide Research Facilities (CURF) at Jeonbuk National University by inductively coupled plasma-mass spectrometry (ICP-MS; 7500A, Agilent Technologies, Germantown, MD, USA).

### 4.5. Sacrifice and Administration

After 12 weeks of the experimental period, the SD rats were sacrificed by anesthetization after 12 h of overnight fasting. Blood was drawn from the abdominal artery, and the serum was separated by centrifugation at 3000 rpm for 15 min at $4\,^\circ$C. Collected epididymal fat and examined the weight without testis. The tissue samples were snap-frozen in liquid nitrogen and stored at $-80\,^\circ$C until analyzed.

### 4.6. Analysis of Serum Biochemical Parameters

Serum total cholesterol (TC), triglyceride (TG), alanine aminotransferase (ALT), and aspartate aminotransferase (AST) levels were analyzed by using commercial kits (Asan Pharmaceutical Co., Seoul, Korea).

The serum concentrations of renin, Ang II, ACE, and aldosterone were analyzed by using enzyme-linked immunosorbent assay (ELISA) kits (MyBioSource, San Diego, CA, USA; Enzo Life Sciences, Farmingdale, NY, USA).

### 4.7. Kidney Histology

Kidney tissue samples were fixed with 10% formalin solution overnight and embedded in paraffin. The tissue samples were then cut into 5-μm-thick sections and stained with hematoxylin and eosin (H&E). Stained areas were viewed using a Leica Microsystems CMS GmbH (Wetzlar, Germany) at $20\times$ magnification, and images were analyzed using SIS 3.2 software (Soft-Imaging System), and glomerular area and glomerular wall thickness were measured using Image J software (National Institutes of Health, Bethesda, MD, USA).

### 4.8. Gene Expression Analysis by Real-Time Polymerase Chain Reaction (RT-PCR)

Total RNA was extracted from kidney and liver tissues using the RNAiso Plus reagent (Takara, Japan). Total RNA was reverse-transcribed into cDNA using a cDNA synthesis kit (Takara, Kusatsu, Japan). PCR was performed in a 7500 Real-Time PCR system (Applied Biosystems, Foster City, CA, USA) using SYBR Green PCR Master Mix (TOYOBO, Japan). Relative gene expression was calculated using β-actin as an internal control. The primer sequences used in this were obtained from PrimerBank. 4.9. Statistical analyses

The results are expressed as the mean $\pm$ standard deviation. Data significant values ($p < 0.05$) were analyzed using one-way ANOVA with SPSS 23 (IBM, New York, NY, USA). Differences among the groups were determined using Duncan's multiple range test (a > b > c > d).

## 5. Conclusions

CTFSS supplementation decreases body weight, epididymal fat weight, and blood pressure in rats, as well as decreased serum renin, Ang II, ACE, aldosterone, and lipid levels, and promotes sodium and potassium ion balance. In addition, the expression levels of adipogenesis- and RAAS-related genes were improved in the tissues. The results of this study suggest that improvement of RAAS-related mRNA levels and adipogenesis mRNA levels might be an underlying mechanism involved in the amelioration of the hypertension by CTFSS. In addition, CTFSS ameliorated hypertension-associated kidney injury. Therefore, further experiments are needed to explore the beneficial effects of soy sauce to beneficial value on hypertension.

**Supplementary Materials:** The following are available online at https://www.mdpi.com/article/10.3390/fermentation7020052/s1, Figure S1: Changes in rat's heart rate during the experiment. Figure S2: Experimental design schedule.

**Author Contributions:** B.Z. completed the experiment, analyzed the results, and wrote the original draft of manuscript; E.-G.M. reviewed and revised the manuscript; J.-X.W. provided assistance; Y.-S.C. was associated with project administration, funding acquisition, and supervision. All authors have read and agreed to the published version of the manuscript.

**Funding:** This work was supported by the National Research Foundation of Korea (NRF) grant funded by the Korean government (MSIT). (No. 2018R1A2B6006477).

**Institutional Review Board Statement:** Institutional Animal Care and Use Committee of Chonbuk National University approved the animal experimental protocol. (CBNU 2016-0030).

**Informed Consent Statement:** Not applicable.

**Data Availability Statement:** The data presented in this study are available on request from the corresponding author.

**Conflicts of Interest:** The authors declare no conflict of interest.

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
