# Peer review of "Chinese Traditional Fermented Soy Sauce Exerts Protective Effects against High-Fat and High-Salt Diet-Induced Hypertension in Sprague-Dawley Rats by Improving Adipogenesis and Renin-Angiotensin-Aldosterone System Activity"

_fermentation, doi:10.3390/fermentation7020052_

Round 1
Reviewer 1 Report
Summary:
Authors investigated effect of Traditional Fermented Soy Sauce on hemodynamic and other mechanisms in experimental model of Male Sprague-Dawley rat during approx. 12 week study and they have found rather favorable effects in all parameters under study compared to high-fat and high fat and salt diet.
Major comments:
Is it possible that the effect of Traditional Fermented Soy Sauce on all parameters was mostly mediated by weight changes? Is it possible that decrease of weight in study group was caused by excessive urination (Table 1 – weight urinary volume in study group was substantially increased, epididymal fat decreased) + do authors have data about amount of stools?
Is it known content of potassium, magnesium in Traditional Fermented Soy Sauce? It could also affect results. Insulin levels and glycemia could add valuable mechanistic information for example regarding insulin resistance; also frequently discussed mechanism of blood pressure and metabolic disorders. I miss information about heart rate. How data regarding epididymal fat were obtained is not described in Methods.
The hypothesis in Introduction and interpretation of Results should be better matched with Discussion – please do not start with lipid metabolism when it was not the main topic of investigation, please start each paragraph by discussion of results from the paper – it is sometimes difficult to get idea which results are discussed – if from this study or references. Some references are not appropriate: Hypertension increases the metabolic burden on the kidneys and causes renal injury [13] – not metabolic burden but inflammatory burden is discussed. That RAAS activation is the main mechanism behind high blood pressure is very simplified statement – in this respect again the reference [20] is not appropriate, the study was focused on non-esterified fatty acids and blood pressure and not RAAS – fatty acids were not analyzed in presented paper.
Minor comments:
Age of models at the end of study is not quite clear, especially duration of particular parts. It could be important – as already proven, 20 to 24 months of age may be the best model for studying the development of systolic hypertension with age (Buñag RD, Teräväinen TL. Tail-cuff detection of systolic hypertension in different strains of ageing rats. Mech Ageing Dev. 1991;59(1-2):197-213.) Some type of timeline, better description of design/duration of experiment in short paragraph could be helpful.
Some expression should be more concise/presented in standard manner: “Body weight and blood pressure administrative “ “increased blood pressure was significantly attenuated compared to …“ Authors should avoid misleading/irrelevant terms as: „ insignificantly different, … insignificant trends ... HDCTS group showed a downward trend, the K+ showed an upward trend, although the difference was statistically insignificant“ Should be clearly stated if changes observed was significant vs. not significant.
Table 1 – statistical results/differences should be more clearly described (a,b,c, . – but what was compared to what is not quite clear). Again not clear statement: “Rats fed with a high-fat and high-salt diet showed significantly higher levels of serum TC, TG, AST, and ALT than those in the HDCTS group, and a significant increase (of what ??) in the HDCTS group compared to the ND and HDS groups.
Fig 4 – Adiponectin.
Table 2 is not be necessary – not too important information.
Conclusion:
Possible changes of weight/excessive urination and loss of fluids as causes of all described effects of Traditional Fermented Soy Sauce should be at least discussed. Another main drawback of this paper are some not correctly cited references. The style of interpretation/discussion of results should be substantially improved.
Author Response
Q1: Is it possible that the effect of Traditional Fermented Soy Sauce on all parameters was mostly mediated by weight changes? Is it possible that decrease of weight in study group was caused by excessive urination (Table 1–weight urinary volume in study group was substantially increased, epididymal fat decreased) + do authors have data about amount of stools?
A1:Thanks for your comment.
Excess weight is known to affect blood pressure rise in human and animal [1]. So, the association of epididymal fat weight with excessive urine excretion as a factor of weight loss was confirmed through Pearson's correlation. As shown in the table below, urine excretion did not correlate with blood pressure and epididymal fat weight. We think it is the weight of epididymal fat that determined the weight changes and gave rise to the weight loss effect.
We are also preparing another manuscript on Chinese soy sauce and obesity. We will reflect your opinion in that paper. In addition, the weight of feces was reflected in Table 1 and line 91.
Body weight |
Urinary volume |
Epididymal fat weight |
Systolic blood pressure |
|
Body weight |
1 |
|||
Urinary volume |
0.390 (0.059) |
1 |
||
Epididymal fat weight |
0.646** (0.001) |
0.181 (0.397) |
1 |
|
Systolic blood pressure |
0.597** (0.002) |
0.110 (0.607) |
0.711** (0.000) |
1 |
**p<0.01 |
[1] AA da Silvar, JJ Kuo, LS Tallam, JE Hall. Role of Endothelin-1 in blood pressure regulation in a rat model of visceral obesity and hypertension. Hypertension. 43:383-387 (2004)
Q2: Is it known content of potassium, magnesium in Traditional Fermented Soy Sauce? It could also affect results. Insulin levels and glycemia could add valuable mechanistic information for example regarding insulin resistance, also frequently discussed mechanism of blood pressure and metabolic disorders. I miss information about heart rate. How data regarding epididymal fat were obtained is not described in Methods.
A2:The sodium and potassium content of traditional soy sauce used in this study was not assessed. However, based on your opinion, we have tried to find the ionic content of Chinese soy sauce, but the database (i.e. Food Data Central) of Chinese soy sauce has not yet been established. Instead, the sodium and potassium content of Korean traditional soy sauce, which has a similar manufacturing process. The Korean traditional soy sauce was more effective in regulating blood pressure than NaCl [2]. Therefore, we expect Chinese soy sauce to have a similar effect.
Besides, our study did not investigate the effect of insulin and insulin resistance on blood pressure control, therefore we have no result to provide in that regards. This study derived results by focusing on RAS control and blood pressure induce by high-fat diet. We will also reflect on this topic in our following study.
We have provided heart rate results as supplementary figure 1, and updated ‘Supplementary Materials section’ (line 323-325).
Also, we have made the correction as per the comment (line 284 to 285).
[2] EG Mun, HS Shon, MS Kim, YS Cha. Antihypertensive effect of ganjang (traditional Korean soy sauce) on Sprague-dawley rats. Nutrition and Research Practice. 11(5):388-395 (2017)
Q3: The hypothesis in Introduction and interpretation of Results should be better matched with Discussion – please do not start with lipid metabolism when it was not the main topic of investigation, please start each paragraph by discussion of results from the paper – it is sometimes difficult to get idea which results are discussed – if from this study or references. Some references are not appropriate: Hypertension increases the metabolic burden on the kidneys and causes renal injury [13] – not metabolic burden but inflammatory burden is discussed. That RAAS activation is the main mechanism behind high blood pressure is very simplified statement – in this respect again the reference [20] is not appropriate, the study was focused on non-esterified fatty acids and blood pressure and not RAAS – fatty acids were not analyzed in presented paper.
A3: We have rearranged the order of the discussions based on your opinion. We corrected incorrect references, sorted the order, and updated the discussion with reference to the results.
We deleted the phrase starting with lipid metabolism in line 162, and started with a sentence about blood pressure. Also, we deleted the existing lines: 51, 226-234 about renal injury, line 64 about RAAS activation, and line 208 about serum aldosterone level.
We updated the line 207 about CVD risk, line 52-55 about renal injuries, and line 67-69 about RAAS.
Q4: Age of models at the end of study is not quite clear, especially duration of particular parts. It could be important – as already proven, 20 to 24 months of age may be the best model for studying the development of systolic hypertension with age (Buñag RD, Teräväinen TL. Tail-cuff detection of systolic hypertension in different strains of ageing rats. Mech Ageing Dev. 1991;59(1-2):197-213.) Some type of timeline, better description of design/duration of experiment in short paragraph could be helpful.
A4: Thank you for your comment. According to the paper you presented, aging increases blood pressure. This study used young rats to eliminate the age variable. As you can see, 24 months of age in rats is equal to 60 years of age in humans. The 24-week-old rats we used corresponded to humans 18 years of age [3]. We wanted to eliminate concerns about increasing hypertension at a young age and check blood pressure control at a young age, where metabolism is active. Based on this study's results, we are preparing research on salt intake and blood pressure control in regards to aging. In the future, our research will be able to prepare a complete answer to this question.
A more detailed experimental design is provided as supplementary figure 2 (line 269), and we have updated ‘Supplementary Materials section’ (line 323-325).
[3] P Sengupta. The laboratory rat: relating its age with human’s. International Journal of Preventive Medicine. 4(6):624-630 (2013)
Q5: Some expression should be more concise/presented in standard manner: “Body weight and blood pressure administrative “ “increased blood pressure was significantly attenuated compared to …“ Authors should avoid misleading/irrelevant terms as: „ insignificantly different, … insignificant trends ... HDCTS group showed a downward trend, the K+ showed an upward trend, although the difference was statistically insignificant“ Should be clearly stated if changes observed was significant vs. not significant.
A5: For a more accurate expression, it has been modified as follows.
We corrected the expression in section 4.3 (line 270)
We deleted ‘increased’ in line 109.
We updated the description about changes observed was significant vs. not significant. (line 98-99)
Q6: Table 1 – statistical results/differences should be more clearly described (a,b,c, . – but what was compared to what is not quite clear). Again not clear statement: “Rats fed with a high-fat and high-salt diet showed significantly higher levels of serum TC, TG, AST, and ALT than those in the HDCTS group, and a significant increase (of what ??) in the HDCTS group compared to the ND and HDS groups.
A6: For a more accurate expression, it has been modified as follows.
We updated the comparison relationship and description of a, b, c, d. (line 103-104, line 115, line 124, line 135, line 149, line 162, and line 312)
We updated the unclear statement in line100-101.
Q7: Fig 4 – Adiponectin.
A7: We apologized for the un-intended spelling mistake at figure 4 about adiponectin, we updated the spelling in figure 4. (line 147).
Q8: Table 2 is not be necessary – not too important information.
A8: We have deleted table 2 and corrected the content in section 4.8(line 307-308).

Reviewer 2 Report
The authors present an original investigation about the effects on adipogenesis of Chinese traditional fermented soy sauce in preventing hypertension in rats. However, there are some issues that need to be address.
In the methods section, the authors explain that the rats were randomly divided into groups however they divided them based on their body weight and similar blood pressure. Therefore, this is not a fully random assignment. Please, give a more detailed description and explain how the body weight and blood pressure was used in the assignment process (e.g., stratification). This is important for the reader to fully understand and value the methodology and results.
In the same section, the authors explain how they measured different parameters, but it is not explained how the epididymal fat weight was measured, although it appears in table 1. As with the other variables. The other variables, the authors should provide a description of how the epididymal fat weight was measured. As this is important to understand the variable and for replicating in further research.
The abstract is lacking a clear description of the methodology. The authors should clearly state that it was an intervention study and if there is space for it, include some information such as the length of the study.
Author Response
Q1: In the methods section, the authors explain that the rats were randomly divided into groups however they divided them based on their body weight and similar blood pressure. Therefore, this is not a fully random assignment. Please, give a more detailed description and explain how the body weight and blood pressure was used in the assignment process (e.g., stratification). This is important for the reader to fully understand and value the methodology and results.
A1: We thank the reviewer for pointing this out. As you said, it was divided appropriately for ‘stratification’ rather than ‘random’.
We are have measured initial body weight and blood pressure in rats after one week of adaptation. And then, we divide into four groups according to body weight and blood pressure with no significant difference. We updated the description in the section 4.2 (line 261-262).
Q2: In the same section, the authors explain how they measured different parameters, but it is not explained how the epididymal fat weight was measured, although it appears in table 1. As with the other variables. The other variables, the authors should provide a description of how the epididymal fat weight was measured. As this is important to understand the variable and for replicating in further research.
A2: We updated the description in the section of 4.5(line 284-285)about epididymal fat weighing
Q3: The abstract is lacking a clear description of the methodology. The authors should clearly state that it was an intervention study and if there is space for it, include some information such as the length of the study.
A3: We updated the description in the abstract about the methodology and length of the study. (line 19)
